# Acceptance and Risk Perception of COVID-19 Vaccination among Pregnant and Non Pregnant Women in Sub-Saharan Africa: A Cross-Sectional Matched-Sample Study

**DOI:** 10.3390/vaccines11020484

**Published:** 2023-02-20

**Authors:** Onyekachukwu M. Amiebenomo, Uchechukwu L. Osuagwu, Esther Awazzi Envuladu, Chundung Asabe Miner, Khathutshelo P. Mashige, Godwin Ovenseri-Ogbomo, Emmanuel Kwasi Abu, Chikasirimobi Goodhope Timothy, Bernadine N. Ekpenyong, Raymond Langsi, Richard Oloruntoba, Piwuna Christopher Goson, Deborah Donald Charwe, Tanko Ishaya, Kingsley E. Agho

**Affiliations:** 1Department of Optometry, Faculty of Life Sciences, University of Benin, Benin City 300283, Nigeria; 2Bathurst Rural Clinical School (BRCS), School of Medicine, Western Sydney University, Bathurst, NSW 2795, Australia; 3Westville Campus, African Vision Research Institute, Discipline of Optometry, University of KwaZulu-Natal, Durban 3629, South Africa; 4Department of Community Medicine, College of Health Sciences, University of Jos, Jos 930003, Nigeria; 5Department of Optometry, Centre for Health Sciences, University of the Highlands and Islands, Inverness IV2 3JH, UK; 6Department of Optometry and Vision Science, School of Allied Health Sciences, College of Health and Allied Sciences, University of Cape Coast, Cape Coast 00233, Ghana; 7Department of Optometry, Faculty of Health Sciences, Mzuzu University, Luwinga 2, Mzuzu P. Bag 201, Malawi; 8Department of Public Health, Faculty of Allied Medical Sciences, College of Medical Sciences, University of Calabar, Calabar 540271, Nigeria; 9Health Division, University of Bamenda, Bambili P.O. Box 39, Cameroon; 10School of Management and Marketing, Curtin Business School, Curtin University, Bentley, WA 6151, Australia; 11Department of Psychiatry, College of Health Sciences, University of Jos, Jos 930001, Nigeria; 12Tanzania Food and Nutrition Center, Dar es Salaam P.O. Box 977, Tanzania; 13Department of Computer Science, University of Jos, Jos 930003, Nigeria; 14School of Health Science, Western Sydney University, Campbelltown, NSW 2560, Australia

**Keywords:** pregnancy, COVID-19 vaccines, acceptance, risk perception, sub-Saharan Africa, misconception

## Abstract

This study aims to evaluate the acceptance and risk perception of pregnant and non pregnant women towards COVID-19 vaccines using a cross-sectional matched-sample study approach. A web-based questionnaire with closed- and open-ended questions was administered to adults older than 18 years in the sub–Saharan African (SSA) region. Respondents (*n* = 131) were grouped based on their pregnancy status (54 pregnant and 77 non pregnant women) and matched for comparison by age. The matched groups were compared using the chi-square test and the *t*-test where appropriate. Compared to non pregnant women, pregnant women reported significantly lower risk perception scores of COVID-19 infection (3.74 vs. 5.78, *p* < 0.001) and were less likely to take the COVID-19 vaccine (odds ratio = 0.12, 95% confidence interval (CI) 0.06–0.27, *p* < 0.001). A similar proportion of pregnant and non pregnant women believed in false information about the COVID-19 vaccine, and 40% of unvaccinated pregnant women (*n* = 40) were concerned about the safety of the vaccine. After adjustment, women’s education, marital status, belief in misconceptions and risk perception were associated with non-vaccination among pregnant women. The content analysis revealed that pregnant women refused the vaccine due to mistrust of their countries’ health systems, concerns about the country where the vaccines were manufactured and a lack of confidence in the production process of the vaccines. This study shows the poor acceptance of COVID-19 vaccines among pregnant women in SSA, who perceived a lower risk of COVID-19 infection. Understanding the reasons for non-acceptance and the motivation to accept the COVID-19 vaccine could guide the development of health education and promotion programmes, and aid governments and policymakers in implementing targeted policy changes.

## 1. Introduction

To reduce the continuous spread of COVID-19, which puts everyone at risk of severe complications and mortality, a large proportion of the population, including pregnant women and children, should be vaccinated [1]. Compared to the general population, pregnant women are at a higher risk of contracting COVID-19, and their overall risk of severe illness from the infection and adverse pregnancy outcomes is greater [1,2,3]. Pregnant women who contract the virus have a higher risk of needing hospitalisation and intensive care [4]. This is partly because pregnancy suppresses the immune response [5] and the growing baby compresses the lungs, causing women to take in less air with each breath [6]. Contracting COVID-19 during pregnancy has also been associated with an increased risk of preterm birth and hospitalisation for the baby [7]. Considering these elevated risks, preventing serious COVID-19 infection is important [8], and various health organisations, including the US Centers for Disease Control and Prevention (CDC), recommend that pregnant women be vaccinated against coronavirus with the assurance of their safety and that of the baby during pregnancy [9,10,11].

Despite the reassurance that vaccination during pregnancy is not associated with any additional pregnancy, birth or new-born complications [12,13,14], many pregnant women are unwilling to be vaccinated due to concerns about the side effects on pregnancy outcomes. These concerns are a result of a lack of data about the safety of the vaccines for the baby and the mother during pregnancy [15,16,17,18]. The results obtained from a qualitative interview of 31 pregnant women across the UK [18] suggested that most participants perceived receiving the vaccine as more dangerous than being infected with COVID-19. Furthermore, the results obtained from a recent review of COVID-19 uptake among pregnant women [19] have revealed that, among over 7000 pregnant women, only 27.5% have been vaccinated against the virus. From their review, the reasons for refusing the vaccines were attributed to a lack of confidence in the government, a confirmed diagnosis during pregnancy and concerns about the vaccines’ side effects and safety. On the other hand, the factors that have been found to improve COVID-19 vaccine acceptability included the woman’s age, race and ethnicity, the fear of being infected with the virus during pregnancy and the trust that the vaccines would prevent them from being infected [19]. However, in the SSA region, one study conducted among pregnant women in northern Nigeria [20] found that primigravid women who are Christian, have a primary level of education, have a higher monthly income, have an earlier gestational age, have received tetanus toxoid in the current pregnancy and have self-assessed their health status as good or better are more likely to accept the COVID-19 vaccine.

Providing adequate information has been suggested as one method of improving COVID-19 vaccine uptake among women [16,21,22,23,24], especially if more safety data on pregnancy become available [25]. Nonetheless, the acceptance rate of COVID-19 vaccines among pregnant women and mothers of young children has been found to differ between geographic locations, with the lowest rates reported for Russia, the USA and Australia. As a result, country-specific vaccination campaigns have been recommended for greater impact [22]. 

There is a paucity of information on the perception of pregnant women about COVID-19 vaccination programmes, particularly among low-income countries and, especially, those in the sub-Saharan African (SSA) region. Considering the low uptake of COVID-19 vaccines among pregnant women, previous findings from a review study suggested different strategies to increase vaccination among pregnant individuals, including promoting evidence-based information on vaccine safety among pregnant women [26]. Therefore, the present study was designed to evaluate the acceptance and risk perception of pregnant and non pregnant women towards COVID-19 vaccines using a cross-sectional matched-sample study approach. The findings of this study may be important to enhance the uptake of already-available vaccine programmes and guide the dissemination of newly developed vaccines. 

## 2. Materials and Methods

### 2.1. Study Design

Existing data from a web-based cross-sectional study carried out between March and May 2021 were analysed for this study. The initial study, designed to evaluate the acceptance of COVID-19 vaccines in SSA, used a convenient sampling method. An e-link to a validated self-administered questionnaire was distributed through e-mails and posted on social media platforms such as Facebook and WhatsApp, inviting participants from all SSA countries, aged 18 and older, to participate. This questionnaire was designed in English and translated into a French version by scholars at the linguistic department of the University of Bamenda, Cameroon, for wider coverage of Anglophone and Francophone SSA countries. 

Initially, 2572 participants (male: 1390 (54.0%); female: 1182 (46.0%)) took part in the study, including pregnant and non pregnant women. For this study, a sample size calculation was conducted. We determined that at least 50 pregnant and 50 non pregnant women were required to detect any statistical differences. In consideration of that, the study had a power of 80% to detect statistical differences, assuming a 10% attrition rate. Subsequently, 54 pregnant women were matched for comparison by age with 77 non pregnant women. Their responses were analysed as illustrated in Figure 1. The distribution of the women by their countries of origin is shown in Figure 2, which indicates that the majority were from Nigeria (32.8%), followed by South Africa (28.2%).

### 2.2. Ethics

Ethical approval was obtained from the Humanities and Social Sciences Research Ethics Committee of the University of KwaZulu-Natal, Durban, South Africa (reference number: HSSREC 00002504/2021). The study adhered to the principles of the 1967 Helsinki Declaration for research involving human participants. An anonymous, voluntary, informed consent was sought from each participant before administering the questionnaire, and participants were instructed to fill out the questionnaire only once. In addition, we ensured single participation from each respondent by utilising IP addresses during analysis. 

### 2.3. Data Collection 

The questionnaire included quantitative and qualitative sections. There were questions to ascertain the respondents’ socio-demographic variables (age group, sex, country of origin, religion, marital status, educational level, employment status, occupational status), knowledge of COVID-19 vaccination and their COVID-19 vaccination status. The questions asked were to determine if participants believed in the efficacy of the vaccines to prevent COVID-19 and its complications and if they had been tested or ever tested positive for COVID-19. The respondents were also asked to indicate if they ‘Agree’ or ‘Disagree’ with the following common misconceptions about COVID-19 vaccines: “COVID-19 vaccines cause infertility in women”, “COVID-19 vaccine is a means to digitally implant microchips” and “COVID-19 vaccines alter DNA”. Other questions included their perception of the risk of becoming infected with COVID-19, the risk of dying from the infection and whether they thought the recommendations for vaccination by the health authorities in their countries were appropriate, with responses on a Likert scale from 0 to 4. The total risk perception score ranged from 0 to 12. 

The vaccination status of the participants (vaccinated and non-vaccinated) was derived from two questions. The vaccinated group responded with an affirmative ‘Yes’ to the question, “Have you been vaccinated against COVID-19?”. The second question was a follow-up to determine the participants’ willingness to get vaccinated when it becomes available in their countries. This question was necessary considering that some SSA countries might not have commenced vaccine distribution to all residents at the time of this study. The responses ‘No’ and ‘Not sure’ to this follow-up question were merged and used to derive the estimate for the non-vaccinated group.

### 2.4. Content Analysis

Two follow-up questions were posed to the non-vaccinated participants, and their responses to these questions were analysed qualitatively. The first question was, “Which of the following factors contributed to your decision not to accept a COVID-19 vaccine?”. For this question, there were ten options, including (1) advice from religious leaders, (2) advice from politicians, (3) mistrust for the pharmaceutical company, (4) mistrust of the health system in my country, (5) mistrust in the medical process for developing the vaccine, (6) mistrust for the country where the vaccine was produced, (7) personal beliefs or past historical experiences with vaccines, (8) concern about the safety of the COVID-19 vaccine, (9) not enough information from healthcare providers and (10) information from the media. 

The second question was, “What can be done to encourage you to take the vaccine?”. For this question, there were eight response options, which included: “I am more likely to accept the COVID-19 vaccine (1) if financial incentives are given to everybody; (2) if monetary rewards are given to healthcare providers involved in the vaccination; (3) if it is given for free; (4) if there is adequate information regarding the specific vaccine; (5) if I can get more education on the vaccines, their side effects, and how effective they are; (6) if it is a travel condition; (7) if it is an employment condition; (8) if many people start receiving the vaccine; (9) if I get positive feedback from those who have been vaccinated”. The open-ended responses were grouped into major codes and analysed. The significant recurrent and salient points were reported using quotations. 

### 2.5. Statistical Analysis

Statistical analysis was conducted using IBM SPSS Statistics for Windows, version 27 (IBM Corp., Armonk, NY, USA). The frequency and percentage of categorical variables were reported. The proportions of vaccinated women who were pregnant, not pregnant and uncertain about vaccination were determined. The association between hesitancy towards the COVID-19 vaccine and the demographic variables was determined using the t-test, the chi-square test and Fisher’s exact test, where applicable. Logistic regression analysis was used to determine the factors associated with COVID-19 vaccination among women in SSA after adjusting for potential confounders. The results were presented as adjusted odds ratios and their 95% confidence intervals. A *p*-value less than 0.05 was considered statistically significant.

## 3. Results

### 3.1. Comparison of Sociodemographic and COVID-19 Test Factors between Pregnant and Non Pregnant Women

The demographic characteristics of the women based on their pregnancy status are shown in Table 1. The majority of the pregnant women were young (18–34 years, 60%), from West Africa, married and had a tertiary education. In contrast, the non pregnant women were spread across three SSA regions, evenly split between two age groups, with the majority being unmarried (83%) and about 48% having a tertiary education. Among the cases and controls, there were predominantly more working women in non-healthcare professions. 

Even though more women agreed that COVID-19 vaccines could prevent COVID-19 infection and its complications, most of them (65% pregnant and 77% non pregnant) non pregnanthad not been tested for COVID-19 infection (Table 1). The results, which are shown in Table 2, revealed that, compared to non pregnant women (23%), a higher proportion of pregnant women (35%) had taken a COVID-19 test at the time of this study, and twice as many pregnant women as non pregnant women had tested positive for the virus (11% vs. 6%). The mean risk perception score determined from the three items in the survey was 3.74 (SD = 2.26) for pregnant women and 5.78 (SD = 2.89) for non pregnant women.

### 3.2. Factors Associated with Non-Vaccination against COVID-19

Table 2 presents the significant variables in the logistic regression. Participants who completed tertiary education, were married, and had the belief that the COVID-19 vaccine is a means to implant digital microchips in one’s body, as well as women who felt at a higher risk of contracting or dying from the virus, were significantly more likely to hesitate or refuse to take the COVID-19 vaccines when they become available in their countries.

### 3.3. Common Misconceptions about the COVID-19 Vaccine

Table 1 also shows the number of pregnant and non pregnant women who held common misconceptions about COVID-19 vaccines. Overall, more women in both groups did not believe the common misconceptions about the vaccine. However, a significant proportion believed that the COVID-19 vaccine alters people’s DNA (79.6% of pregnant women and 76.6% of non pregnant women). Approximately half of the pregnant women and 40% of the non pregnant women believed that the vaccine causes infertility. These beliefs were not dependent on the vaccination status of the participants.

The percentage of pregnant women and their past vaccinations is depicted in Figure 2. Overall, a higher proportion of pregnant women reported having been vaccinated in the past for other conditions compared to non pregnant women, especially against yellow fever (57% vs. 42%) and polio (54% vs. 43%). 

In the univariate analysis, there was a significant difference in the likelihood of receiving the COVID-19 vaccines between pregnant and non pregnant women (odds ratio: 0.12, 95% Cl: 0.06–0.27). At the time of this study, 26% of pregnant women and 74% of non pregnant women had been vaccinated against COVID-19 (Figure 3).

### 3.4. Reasons for Not Getting Vaccinated against COVID-19

Figure 4 presents the breakdown of the reasons for not getting the COVID-19 vaccine among unvaccinated pregnant women. The most frequently cited reason by pregnant women for not taking the COVID-19 vaccines was mistrust of the health system in their countries (*n* = 19), while others (*n* = 16) cited the safety of the vaccines as their main reason for not receiving them. Information from the media and advice from religious leaders contributed the least to the reasons why pregnant women were hesitant towards the COVID-19 vaccines (*n* = 5), whereas the views of politicians about the vaccines did not influence the women’s decision regarding COVID-19 vaccination.

Participants were also asked to indicate other reasons why they were not vaccinated. This was an open-ended question. Figure 5 presents the common themes that emerged as reasons for not being vaccinated. Apart from a few pregnant women who indicated that the unavailability of vaccines contributed to their not being vaccinated, most women who had not received the COVID-19 vaccine said it was mainly because of their suspicions about the countries where the vaccines were produced and the uncertainty of the vaccine production. Others reported concerns about the safety of the vaccines as the main reason for not taking them at the time of this study.

Below are some of the quotations from the women who said they would not take the COVID-19 vaccine when it became available to them:


*“Concerned about the effects after taking the vaccine. There are many myths concerning it, like, it can make a woman not fertile to depopulate us. Most importantly, our COVID strain in Africa is not that dangerous. They should make available the vaccine to be given to the developed countries like us and not another product”.*



*“I rather prefer self-protection for prevention purposes than trust the vaccine”.*



*“Personal conviction that the vaccine is not necessary for Africa, especially for young people who are not at risk. It could be a birth control procedure to reduce world population”.*



*“Vaccines have been used against black people for far too long-Kenya infertility, Tuskegee, etc. This vaccine is as questionable and its benefits for politicians far outweigh its care to manage this self-limiting bug”.*


Many respondents stated that they did not take the vaccines due to a dearth of information from healthcare providers about them. Others, however, said they refused the vaccines following advice from their religious leaders and their personal beliefs. Lastly, others reported that it was out of fear from their experience with other vaccines and their health, as can be seen from the quotes below:


*“Risk to my health as I have SLE with a severely compromised immune system”.*



*“I have a diagnosed allergy, which is the main cause of asthma and skin reactions, conjunctivitis. I am scared I might react to the vaccine”.*


### 3.5. Motivations to Get COVID-19 Vaccines

About one-quarter of the participants accepted that they would take the vaccine if it were made available to them. Other participants indicated an unwillingness to take the vaccine, while some of the participants were uncertain about their willingness to take the vaccine. One of the participants clearly stated that the reason she was unvaccinated was that the “Government has not just vaccinated the mass population”. 

The majority said they would accept the vaccine if more information were to be provided about the production, availability, safety and side effects of the COVID-19 vaccine, while a significant number also said they would accept the COVID-19 vaccine only if it were given for free of charge or if it were a condition for travelling. 

Other participants said that they would accept the vaccine if they were given some form of incentive. Pregnant women were more concerned with feedback about their health and the health of their unborn babies. The following responses typify this: “if I get more education on the vaccines, their side effects and how effective they are” (51.9% cases, 37.7% controls); “if I get positive feedback from those vaccinated” (51.9% cases, 29.9% controls). On the other hand, non pregnant women were more concerned about travel conditions (16.7% cases, 27.3% controls), employment and financial inducements. The participants’ responses regarding the reasons that could increase vaccine acceptance are presented in Figure 6.

## 4. Discussion 

This study compared the uptake of COVID-19 vaccines among pregnant and non pregnant women in SSA, who were matched by age. For the non-vaccinated pregnant women, including those who were hesitant or did not intend to take the COVID-19 vaccines when they became available in their countries of residence, we also determined the reasons for their decisions and identified the factors associated with hesitancy and refusal to take the vaccine. Multivariable analysis revealed that level of education, marital status, belief in the common misconception that the vaccine was meant to implant a microchip into the body and higher risk perception were significantly associated with non-vaccination against COVID-19 in this study. 

Despite having received previous vaccinations for other conditions, pregnant women were significantly less likely to take the COVID-19 vaccines compared to non pregnant women in this study, which is likely to increase their risk of severe complications if infected. For those women who indicated they had not had access to the vaccine yet, one main reason could be the reduced availability of the vaccines in Africa [27]. Interestingly, it can be seen from this study that more pregnant women took the COVID-19 vaccine compared to the flu vaccine (about 30% vs. less than 20%). A similar report was given in a retrospective study [28], where just under 20% of pregnant women out of about 500,000 got vaccinated against influenza. The fact that influenza is not easily differentiated from other rampant infectious diseases (presenting with fever), such as malaria, which occur in the tropics [29], may have accounted for less attention being paid to this vaccination. 

From this study, it was observed that pregnant women had a higher proportion of those who had tested (and were) positive for COVID-19. However, they had the lowest proportion of those who were vaccinated, despite being at higher risk. The higher proportion of pregnant women who had taken the COVID-19 test could be due to concerns about not wanting to be infected, or they may have been asked to take the test by their healthcare providers. The finding that twice the number of pregnant women tested positive may have resulted from more pregnant women having access to these tests. Furthermore, the low acceptance of vaccines among pregnant women was also found in a study conducted in northern Nigeria [20], where only one-third of the respondents indicated that they would accept the vaccine during pregnancy. The low vaccine acceptance found in the present study may be associated with the safety concerns expressed by the women since most believed in common myths about the COVID-19 vaccines, which significantly influenced the low uptake. This is not different from what was found in other studies, especially among Africans, where concerns about the safety of the vaccines were the reasons for vaccine hesitancy [30]. Notwithstanding, some side effects have been reported, mostly mild and expected, such as pain at the site of injection, headache and, in some rare cases, allergic reactions [31].

The findings of this study showed that pregnant women had a lower perception of the risk of getting infected and dying from COVID-19. This may suggest that they were unaware of the implications of being infected with the coronavirus disease while pregnant. Lack of information was also part of the reasons given for non-vaccination in this study. A higher perception of the risk of a disease ordinarily leads to greater compliance with health measures. Issues of health and safety concerns were more paramount for pregnant women, as revealed by their responses to the reasons that could increase their vaccine acceptance. 

The safety of the vaccines, which most pregnant women agreed was an issue, portrays similar findings to a previous study [18] where respondents knew that infection with the virus could be potentially fatal but refused to take the vaccines due to doubts about their safety for themselves and their unborn children. This finding, therefore, highlights the importance of proper vaccine education to increase acceptance. 

The responses to factors that encourage COVID-19 vaccine acceptance further identified pregnant women as very concerned about the safety of the vaccines. Of all the conditions asked, the responses with the highest percentage were related to the effectiveness and safety of the vaccines. Additionally, more pregnant women lacked trust in the health system of their countries. In a systematic review, authors found that factors such as trust in the safety and efficacy of vaccines, trust in the individuals who administer or give advice about the vaccines and trust in the healthcare systems of countries are all important in the vaccine decision-making process [32]. The lack of trust observed in this study is not far from their lack of confidence in the ability of the health system to appropriately manage their condition when a problem or complication arises due to the deplorable state of most health facilities in Africa and their concern that health professionals lack the required competence to handle the novel disease. The emergence of COVID-19 exposed the poor conditions of health systems in terms of infrastructure, equipment, drugs and human resources required for standard patient care. Additionally, the history of mistrust from past interactions with official institutions may have influenced the public trust of the participants in this study. Such diverse histories and experiences may lead to highly variable and locally specific public trust in vaccines and other immunisation programmes in society [33].

A recent study [34] that evaluated the functioning of the health system in SSA, including challenges and responses, identified the poor structure of health systems and a dearth of essential health services as major setbacks during the COVID-19 pandemic. These weaknesses, coupled with the unmet demands arising from the COVID-19 pandemic, may have contributed to the mistrust of pregnant women towards the health care system. Being eager to receive positive feedback from others highlights the need to constantly educate women so they can make informed choices [35]. A detailed record of vaccine dissemination and outcomes may also be needed to aid this education.

### Limitations and Strengths

Vaccination campaign programmes could be designed based on the results of this study, particularly considering the participants’ intention to vaccinate. However, there are some limitations to this study, including the convenient sampling of online users and women in rural areas with limited internet access, which limits the generalizability of our findings beyond the study sample. This is important, considering that online users were more likely to believe the common myths about the COVID-19 vaccine that could potentially reduce vaccine uptake among women [34]. In addition, key indicators such as the postpartum period and parity were not investigated because the study was not specifically designed for pregnant and postnatal women. The study also did not investigate whether the COVID-19 vaccines were available in the countries during the survey and, therefore, the participants’ decisions might change whenever the vaccines became available later. However, at the time of this study, some African countries had either just rolled out the vaccination programme [36,37] or targeted only front-line health workers [38]. Despite these limitations, the strength of this study is in the mixed-method approach, which provided more insight into the perception of pregnant women on vaccine hesitancy and reasons for non-vaccination among this high-risk group in the SSA region. Second, the language diversity of both the English and French versions of the survey also captured opinions from members of Francophone and Anglophone countries spanning 17 countries in SSA. Third, the robustness of the analysis minimised the influence of potential confounders. Lastly, this study used a validated questionnaire shown to have satisfactory internal validity among SSA respondents [39]. However, further studies targeting pregnant women are needed in the region to provide an in-depth analysis of the reasons behind their decisions regarding COVID-19 vaccine uptake and the influence of social media.

## 5. Conclusions

This study has shown that over two-thirds of pregnant and non pregnant women in SSA agree that COVID-19 vaccines can prevent COVID-19 infection and its complications. However, only one in four pregnant women was vaccinated, despite their higher rate of previous vaccinations. The lower vaccination rate could be attributed to their lower perceived risk of being infected, their greater likelihood of believing in the false information about the COVID-19 vaccine and their increased concern about the vaccine’s safety, in addition to the mistrust of their countries’ health systems and their lack of confidence in the production process of the vaccines. More enlightenment campaigns should be carried out to create awareness about the safety of the vaccines, primarily targeted at high-risk groups, to emphasise the safety and efficacy of the COVID-19 vaccine, as well as dispel any misconceptions regarding common false beliefs. Public health officials can also seize this opportunity to establish meaningful relationships with the communities they serve to gain their trust, which may in turn increase the uptake of the COVID-19 vaccination. These approaches should target women who are married, have tertiary education and have a high perception of the risk of contracting the virus. Most importantly, this information is crucial for governments and policymakers to make targeted policy changes for future pandemics.

## Figures and Tables

**Figure 1 vaccines-11-00484-f001:**
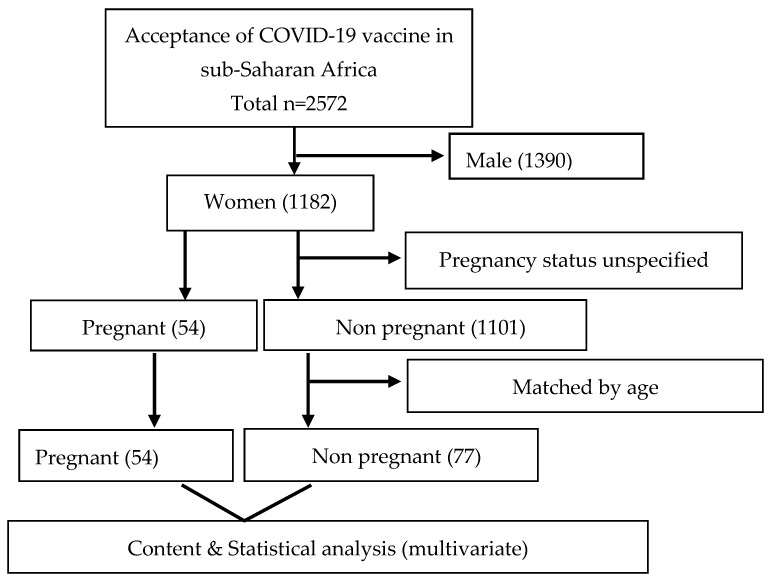
Flowchart of pregnant and non pregnant women in sub-Saharan Africa.

**Figure 2 vaccines-11-00484-f002:**
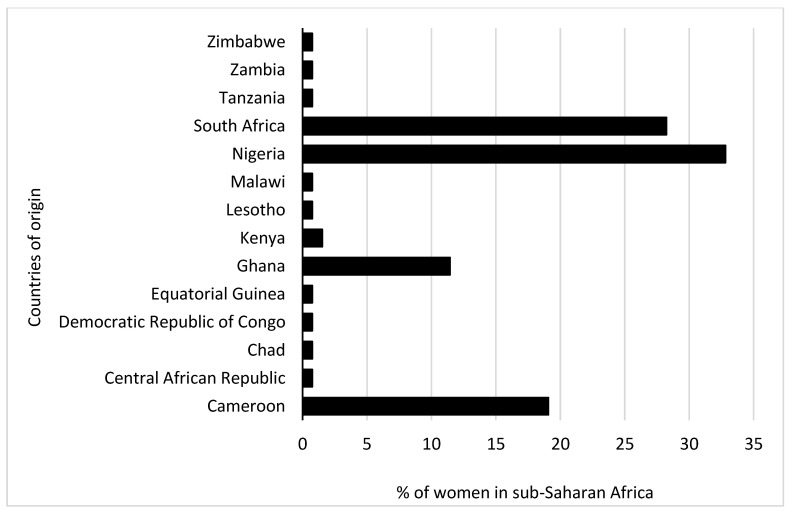
The proportion of women in sub-Saharan Africa by country of origin.

**Figure 3 vaccines-11-00484-f003:**
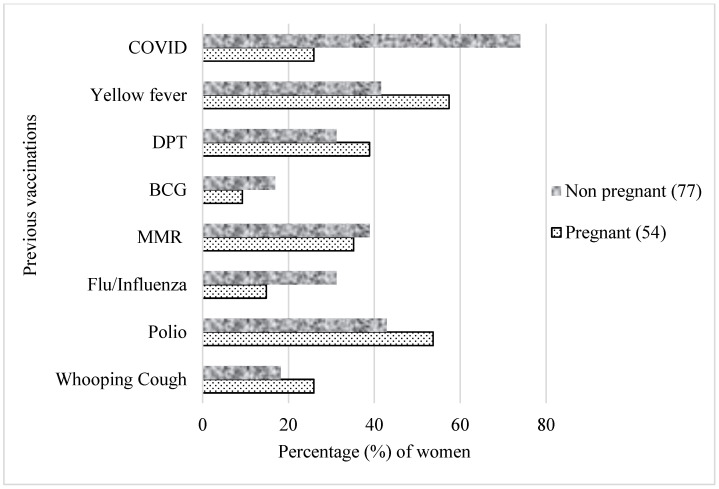
Previous vaccinations based on pregnancy status. Participants selected multiple responses. MMR—measles–mumps–rubella combination vaccine; DPT—diphtheria, pertussis and tetanus; BCG—Bacille Calmette–Guérin.

**Figure 4 vaccines-11-00484-f004:**
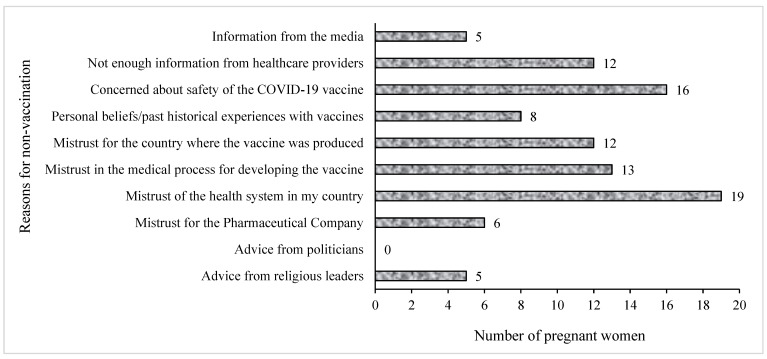
Reasons for non-vaccination against COVID-19 among pregnant women. participants selected multiple responses.

**Figure 5 vaccines-11-00484-f005:**
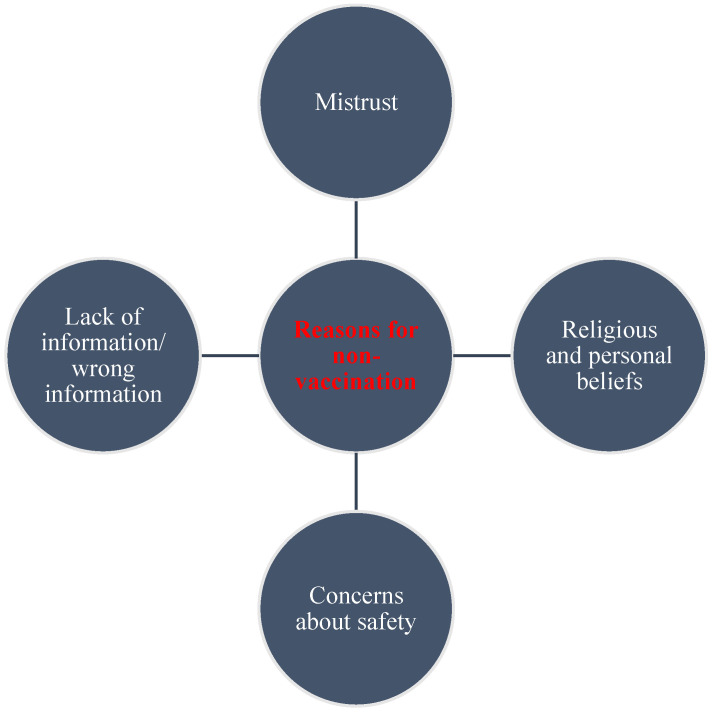
Emergent themes for reasons for rejecting the COVID-19 vaccines.

**Figure 6 vaccines-11-00484-f006:**
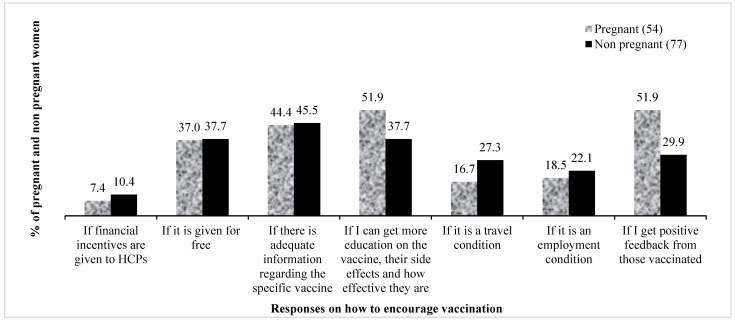
Conditions that would encourage acceptance of the COVID-19 vaccines. HCPs—healthcare practitioners.

**Table 1 vaccines-11-00484-t001:** Socio-demographic and COVID-19 testing among pregnant and non pregnant women.

Variable	Pregnant Women (n = 54, 41.2%)	Non Pregnant Women (n = 77, 58.8%)	*p*-Value
**Demography**			
**Region of origin**			
West Africa	30 (56.6)	28 (36.36)	0.037
East Africa	4 (7.55)	2 (2.60)	
Central Africa	8 (15.09)	20 (25.97)	
Southern Africa	11 (20.75)	27 (35.06)	
**Age**			
18–34 years	32 (60.38)	35 (50)	0.252
35 and older	21 (39.62)	35 (50)	
**Marital status**			
Unmarried	15 (27.78)	64 (83.12)	<0.001
Married	39 (72.22)	13 (16.88)	
**Education**			
Tertiary	50 (92.59)	37 (48.05)	<0.001
Secondary	4 (7.41)	40 (51.95)	
**Employment status**			
Unemployed	14 (25.93)	28 (36.36)	0.208
Employed	40 (74.07)	49 (63.64)	
**Occupation**			
Non-healthcare worker	36 (66.67)	59 (76.62)	0.209
Healthcare worker	18 (33.33)	18 (23.38)	
**Place of residence**	*n = 53*		
Africa	52 (98.11)	73 (94.81)	0.335
Diaspora	1 (1.89)	4 (5.19)	
**COVID-19 test factors**			
**COVID-19 vaccine can prevent COVID-19 infection and its complications**			
Disagree	11 (20.37)	26 (33.77)	0.492
Agree	43 (79.63)	51 (66.23)	
**Have you ever been tested for coronavirus disease (COVID-19)?**			
No	35 (64.81)	59 (76.62)	0.139
Yes	19 (35.19)	18 (23.38)	
**Have you ever tested positive for coronavirus disease (COVID-19)?**			
No	48 (88.89)	72 (93.51)	0.348
Yes	6 (11.11)	5 (6.49)	
**Common misconceptions about the COVID-19 vaccine**			
**COVID-19 vaccines cause infertility in women**			
Disagree	29 (53.70)	46 (59.74)	0.492
Agree	25 (46.30)	31 (40.26)	
**COVID-19 vaccine is a means to digitally implant a microchip**			
Disagree	31 (57.41)	53 (68.83)	0.094
Agree	23 (42.59)	24 (31.17)	
**COVID-19 vaccines alter DNA**			
Disagree	11 (20.37)	26 (33.77)	0.180
Agree	43 (79.63)	59 (66.23)	
**Perception of risk of COVID-19 infection**			
Mean (SD)	3.74 (2.26)	5.78 (2.89)	<0.001

Tertiary = Diploma, university or postgraduate degree; unmarried = widowed, divorced, separated or single.

**Table 2 vaccines-11-00484-t002:** Multiple logistic regression analysis of factors associated with non-vaccination among pregnant women in sub-Saharan Africa.

Variable	AOR [95%CI]	*p*-Value
Education		
Tertiary	1.00	
Secondary	0.04 [0.01, 0.18]	<0.001
Marital status		
Unmarried	1.00	
Married	37.54 [9.30, 151.56]	<0.001
COVID-19 vaccine is a means to implant a digital microchip		
No	1.00	
Yes	3.63 [1.12, 11.79]	0.032
Perception of risk of COVID-19 infection	1.58 [1.24, 2.01]	<0.001

AOR—adjusted odds ratio; CI—confidence interval; COVID-19—coronavirus 2019.

## Data Availability

The data presented in this study are available on request from the corresponding author. The data are not publicly available due to institutional policy. Data can be made available on request from the corresponding author.

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
