# Peer review of "Acceptance and Risk Perception of COVID-19 Vaccination among Pregnant and Non Pregnant Women in Sub-Saharan Africa: A Cross-Sectional Matched-Sample Study"

_vaccines, 2023, doi:10.3390/vaccines11020484_

Round 1

Reviewer 1 Report

The study is important, but the paper is sometimes hard to read. It would benefit from breaking sentences down. 

One of the most critical points is that the authors assess refusal of the  vaccine for women who did not yet have access to the vaccine. thus, they did not refuse, rather they did not intend to take it. these are quite different and the analysis should be separated for those who had access to the vaccine and those who did not yet have access. 

figure 2 would benefit from including covid vaccine, so we can easily see the differences in acceptance. but you still need to account for those who do not have access, as they are not able to accept or reject it.

Mistrust in the HCsystem appears to be important. so, it would be helpful to add to the discussion why people mistrust the system, what kind of HC system and governance system these women live in. maybe the mistrust is not isolated to vaccines?

and this statement is completely false:

A previous study [28] has shown that other vaccine intake (human papillomavirus, varicella, measles/mumps/rubella, and anthrax vaccines) during pregnancy could indeed cause a major defect in the infants born afterwards, ranging from the musculoskeletal, nervous, circulatory, urinary, digestive, and circulatory systems, including defects in the eyes, ears and even chromosomal abnormalities. Therefore, it is not surprising that this fear of COVID-19 vaccine intake exists among some pregnant women.

WRONG – the study found that BD were INFREQUENT. And, the conclusion that women in SSA know this study from USA, published in "Human Vaccines & Immunotherapeutics" is a stretch.

Author Response

Response to reviewers’ comments

Dear Editor

We appreciate the opportunity to have our article published in your journal. We have provided a point-by-point response to the reviewer’s comments. In the manuscript, we also made changed which were highlighted in red for the responses to reviewers comments and in blue for changes made during editing by an English speaking person.

Reviewer 1

Comments and Suggestions for Authors

  1. The study is important, but the paper is sometimes hard to read. It would benefit from breaking sentences down. 

Response: Thank you for the comment. The paper has been revised and long sentences have been broken down into short ones

  1. One of the most critical points is that the authors assess refusal of the vaccine for women who did not yet have access to the vaccine. thus, they did not refuse, rather they did not intend to take it. these are quite different, and the analysis should be separated for those who had access to the vaccine and those who did not yet have access. 

Response: Thank you for this suggestion. The study did not investigate whether the COVID vaccine was available in the countries during the survey, so we are therefore unable to separate the participants who had access to the vaccine and those who did not yet have access. This has been included as a limitation of the current investigation. Page 11, lines 361 - 363 ‘Again, the study did not investigate whether the COVID vaccines were available in the countries during the survey and therefore the participants’ decisions might change whenever the vaccines became available later’

However, from the participants’ responses to the survey question, ‘would you be willing to take the vaccine, if available in your country?’ we derived the variable refusal if they responded with a ‘No’ and hesitancy if they responded with ‘not sure’. Even though their decision may change over time, this was the best estimate of refusal that we could deduce from the data. For this reason, no change was made in the context of refusal as used in the manuscript.

  1. figure 2 would benefit from including covid vaccine, so we can easily see the differences in acceptance. but you still need to account for those who do not have access, as they are not able to accept or reject it.

Response: Thank you for the suggested comment. Fig 2 has been revised to include those who had COVID vaccine

  1. Mistrust in the HC system appears to be important. so, it would be helpful to add to the discussion why people mistrust the system, what kind of HC system and governance system these women live in. maybe the mistrust is not isolated to vaccines?

Response: The authors thank the reviewer for this useful comment. We have added the following paragraph in the discussion:

“….. and a lack of confidence in the health care system. A recent study [29] which evaluated the health system functioning in SSA including the challenges and responses has identified the poor structure of health systems and a dearth of essential health services as major setbacks in the midst of the COVID-19 pandemic. These weaknesses coupled with the unmet demands arising from the COVID-19 pandemic may have accounted for the pregnant women’s mistrust of the health care systems. ”[lines 359-366]

  1. and this statement is completely false:

A previous study [28] has shown that other vaccine intake (human papillomavirus, varicella, measles/mumps/rubella, and anthrax vaccines) during pregnancy could indeed cause a major defect in the infants born afterwards, ranging from the musculoskeletal, nervous, circulatory, urinary, digestive, and circulatory systems, including defects in the eyes, ears and even chromosomal abnormalities. Therefore, it is not surprising that this fear of COVID-19 vaccine intake exists among some pregnant women.

WRONG – the study found that BD were INFREQUENT. And, the conclusion that women in SSA know this study from USA, published in "Human Vaccines & Immunotherapeutics" is a stretch.

Response:  Thanks for the observation. We have reviewed the manuscript and included a revised reference. See page 11, lines 323 – 329. The revised section now reads as follows:

“The low COVID vaccine acceptance found in the present study may be associated with the safety concerns expressed by the women most of who believed in the common myths about the COVID vaccines. A previous study [28] also showed that birth defects among infants born to women who took other vaccines (human papillomavirus, varicella, measles/mumps/rubella, and anthrax vaccines) during pregnancy occurred, though infrequent.

Reviewer 2 Report

Dear Authors, please see the attached file

Author Response

Review Vaccines 2186663
Major revisions
Lines 81-83. These lines should be removed since they refer to a single study with a few participants. Instead, you should use the information from systematic reviews and meta-analyses among pregnant women, such as 10.3390/vaccines10050766.
Response: Done. See lines 88-101
Please, expand the Introduction section adding more information regarding the factors that affect acceptance and perception of COVID-19 vaccination among pregnant and non-pregnant women especially in sub-Saharan Africa.
Response: We have expanded on the introduction. Same lines 88-101
Lines 99-101. I cannot understand this sentence. Since you have performed a large survey on the same topic, what the present manuscript adds to the literature? Also, you do not report the reference for the large survey.
Response: This section was revised because the paper being referred to is still unpublished at this time[lines 111-115]. It now reads like so:
“Considering the low update of COVID vaccines among pregnant women, previous finding from a review study suggested different strategies for increased vaccination within pregnant individuals including promoting the evidence-based information on vaccine safety among pregnant women[reference]”
How did you validate your questionnaire and especially the perception score?
Response: The study used a validated questionnaire shown to have satisfactory internal validity among SSA respondents especially with respect to the risk perception scores (Osuagwu UL, et al. International Journal of Environmental Research and Public Health. 2021 Oct 21;18(21):11091.).
Sample size needs more explanation. What is the type I error?
Response: we have expanded on the section on sample size. It now reads:
“For this study, a sample size calculation was conducted. We determined that at least 50 pregnant and 50 non-pregnant women were sufficient to detect any statistical differences, and with that, the study had a power of 80% to detect statistical differences considering a 10% attrition rate. Subsequently, 54 pregnant women were matched for comparison by age, with 77 non-pregnant women and their responses were analyzed as illustrated in Figure 1”
What is the level of statistical differences that you detect (low, medium, or high)? What measure of association did you use?
Response: These have been shown in the statistical analysis section and now reads:
“The association between hesitancy towards the COVID-19 vaccine and the demographic variables was determined using the t-test, chi-square test and Fisher’s exact test for small number of cells, where applicable A P value less than 0.05 was considered statistically significant.”
Since you performed a matched study, why did you use 77 non-pregnant women instead of 54 as were the pregnant women.
Response: We had unequal number of women in both groups because the participants were matched by age and not a a 1:1 matching which is often done in case control studies.
Which countries from SSA were participating in your study? Which are the numbers from each country?
Response: The distribution of the participants by countries of origin is now shown as Figure 2
There is a confusion regarding the qualitative form of your study. Since you did not perform interviews with the participants and you used questionnaire with closed-ended questionnaire, your study is clearly a quantitative one. You can present the open-ended comments for one question using numbers and percentages. Please, re-write this Methods section (lines 161-182). Similarly, you should not use quotations from the women since you did not take interview from them (lines 263-284). You should re-write these results.
Response: Contrary to the reviewer’s comments, the items used to derive the qualitative section utilised an open-ended question. This has been made clearer in Lines 260-261 and now reads:
“Participants were also asked to indicate other reasons why they were not vaccinated. This was an open-ended question. These responses were analysed to derive themes as described in the methods”. In addition, these were quotations from the women and we have retained the quotation marks.
You performed a matched study according to age, marital status and education but there are great differences between the two groups regarding the matching variables. 93% of pregnant women had tertiary education while the respective percentage for non-pregnant women was 48%. Similarly situation for age and marital status. How this could happen?
Response: Thanks for picking this up – it was a typo and we have revised the manuscript to reflect the actual matching method we implemented.
Please, perform a univariate and multivariable logistic regression analysis to assess the impact of independent variables on women acceptance and perception of COVID-19 vaccines.
Response: Univariate analysis was already conducted (Lines 240-242). However, multivariable logistic regression analysis is not needed because we have matched our data and methods for controlling for confounders are multivariate analysis, randomization, restriction, matching, and stratification (Jager et al., 2008; Greifer & Stuart, 2021).
Reference:
Jager, K. J., Zoccali, C., Macleod, A., & Dekker, F. W. (2008). Confounding: what it is and how to deal with it. Kidney international, 73(3), 256-260.
Greifer, N., & Stuart, E. A. (2021). Matching methods for confounder adjustment: an addition to the epidemiologist’s toolbox. Epidemiologic reviews, 43(1), 118-129.
Discussion section is limited, and you should expand it. For example, you only mention two references while there is a great amount of the literature regarding the reasons that pregnant women accept or not COVID-19 vaccination. After the multivariable logistic regression analysis, you should discuss your findings.
Response: The discussion has been expanded and few new references added. These are highlighted in red and blue.
You mention sample size as appropriate according to your calculations but then you report that it is a limitation of your study. Also, sample size does not limit the generalizability of your findings but your study design. Please, further discuss it. Moreover, there is plenty of other bias in your study: selection bias due to a convenience sample with on-line data collection; information bias, use of a non-reliable and valid questionnaire, pregnant women who did not have the opportunity to take a COVID-19 vaccine, etc.
Response: Agreed, like many cross sectional web-based survey, the study limitations are inherent and we have expanded on them in the limitation [Lines 385-396]
Minor revisions
Please see the order of the references throughout the text. Reference 26 is in line 102 and the next reference is with the number 49 in line 160.
Response: Thanks for pointing this out. Both references have been revised following other reviewer’s comments. Reference 26 has been cited properly and reference 49 deleted because it was wrongly cited.
Please, remove Table 2 and Figure 3 since they present only a few numbers.
Response: Table 2 was deleted, a new figure 2 was made from your comments. The previous Fig 2 is now Fig 3 and now includes the COVID vaccination as requested by Reviewer 3. Other figures were retained for easy visualization of the reasons for non vaccination.
Line 5. Please remove the title from the Author.
Response: Done
Line 45. Before the abbreviation, please write the full term.
Response: Done
Line 50-52. Please add measures of effect and p-value or/and confidence intervals.
Response: This section was revised based on revised findings.
It now reads:
“A similar proportion of pregnant and non-pregnant women believed in the false information about the COVID-19 vaccine and 40% of the unvaccinated pregnant women (n=40) were concerned about the safety of the vaccine”
Figure 1 does not show all the matching variables.
Response: Figure 1 shows the selection of participants matched by age. This has been revised to indicate the matching that was performed.
English narrative needs to be firmly corrected, please check spelling, grammar errors, and wordiness across the manuscript. Typos, and grammatical mistakes need fixing
Response: The manuscript has been revised by a native English-speaking person.

Reviewer 3 Report

This is an interesting article on acceptance and perception of COVID-19 vaccination among pregnant and non-pregnant women in sub-Saharan Africa that was investigated by a cross-sectional matched sample study.

This study has shown poor acceptance of COVID-19 vaccines among pregnant women in SSA and perception of lower risk of COVID-19 infection. The reasons for non-acceptance and motivation to accept the COVID-19 vaccine emerged from this study could guide the design of health education and promotion programs, and support governments and policymakers for targeted policy changes in view of future pandemics.

The structure and organization of the article appear in line with the journal ‘s instructions.

Minor changes are indicated in the text.

Author Response

Response: Thanks for the comments

Minor changes are indicated in the text.

Revise this sentence:

However, a large Israeli study [20] which compared the safety of the Pfizer/BioNTech vaccine between women who received one or two doses of the vaccine during pregnancy and unvaccinated women, found no difference in preg-86 nancy, delivery and newborn complications between the group for gestational age at de-87 livery, including the incidence of small for gestational age and newborn respiratory com-88 plications.

Response: Thank you for the comment. The sentence has been revised to read as follows (page 2, lines 82 - 86):

“However, Wainstock et al [20] found no difference in pregnancy outcomes, delivery and newborn complications of Israeli women who received either one or two doses of the Pfizer/BioNTech vaccine and those who did not receive the vaccine”

Revise this sentence:

This study compared the vaccine uptake rate among pregnant and age and regional-307 matched a non-pregnant woman in SSA, their reasons for non-vaccination and the factors 308 associated with non-vaccination

Response: Done. We have broken down the sentence into two

“This study compared the uptake of COVID vaccines among pregnant and non-pregnant women in SSA who were matched by age and region. For the non-vaccinated pregnant women, including those who were hesitant or did not intend to take the COVID vaccines when they became available in their countries of residence, we also determined the reasons for their decisions, as well as identified the factors associated with hesitancy and refusal to take the vaccine”

Round 2

Reviewer 1 Report

the authors of this paper did the minimal effort to address my concerns and in some cases did not do so adequately.

re point 4:
they simply added a few lines about "maybe people mistrust the system", when they need several paragraphs about why trust in health systems influences trust in medicines and vaccines in particular, what type of health systems people in the study live in, what levels of trust do people have in those systems, what has happened historically that could influence trust in covid vaccines? they simply ignore a variable that is key to the study.

“….. and a lack of confidence in the health care system. A recent study [29] which evaluated the health system functioning in SSA including the challenges and responses has identified the poor structure of health systems and a dearth of essential health services as major setbacks in the midst of the COVID-19 pandemic. These weaknesses coupled with the unmet demands arising from the COVID-19 pandemic may have accounted for the pregnant women’s mistrust of the health care systems. ”[lines 359-366]
this does not address my concern adequately.

re point 5
they simply rephrased the statement ever so slightly instead of reviewing a broader literature on side effects, adverse events, media coverage of adverse events, or -- a very easy and appropriate thing to do would be to -- cite recent papers about reasons for covid vaccine hesitancy or refusal. they tried to "soften" their claim about birth defects when that paper - published too long ago - shows they are RARE. I would remove the senetence and reference to that paper, as it is not serving the hypothesis. since so many people in the study mistrust the health system, mistrust the country where the vaccine is made, and are concerned about saftey, the authors need to focus on literature that could explain why they have such low trust. instead, they are spreading reasons to mistrust based on 1 old paper that found birth defects to be rare (a postive vs a negative). 
in the conclusion they state: dispel any misconceptions regarding common false beliefs. but i think they add to the mis-information in this claim.

point 2: they claim in the introduction that some women did not have access to covid vaccines yet. but they show no evidence. "they did not assess it" is not an adequate answer. they can easily look at the vaccine rollout of the countries where the survey participants come from and then they know how big or small this issue is. They have the IP address of participants, so while that may not be a perfect match and some may have used VPN, it does give some insights into the availability of the vaccine for participants

conclusion should be focused on the aim of the paper and the data. it now seems "off". it does not mention anything about trust building. if people mistrust at the levels they do, why is the conclusion that health promotion officers should educate these women? people don't trust them.

this study has great potential to fill a gap in the global literature, as they state is important in the intro (and I agree that it is), yet it falls short in the discussion and conclusions. how is SSA similar or different that other parts of the world when it comes to women (pregnant or not) accepting coivd vaccines?

the fact that many more pregnant women got the covid-19 vaccine  (~30 vs <20%) when compared to the flu vaccine is worth discussing.

Author Response

Reviewer 1

Comments and Suggestions for Authors

the authors of this paper did the minimal effort to address my concerns and in some cases did not do so adequately.

re point 4:
they simply added a few lines about "maybe people mistrust the system", when they need several paragraphs about why trust in health systems influences trust in medicines and vaccines in particular, what type of health systems people in the study live in, what levels of trust do people have in those systems, what has happened historically that could influence trust in covid vaccines? they simply ignore a variable that is key to the study.

“….. and a lack of confidence in the health care system. A recent study [29] which evaluated the health system functioning in SSA including the challenges and responses has identified the poor structure of health systems and a dearth of essential health services as major setbacks in the midst of the COVID-19 pandemic. These weaknesses coupled with the unmet demands arising from the COVID-19 pandemic may have accounted for the pregnant women’s mistrust of the health care systems. ”[lines 359-366]
this does not address my concern adequately.

Response:

Thanks for the suggestions. We have expanded on this section. The revised section now reads:

Responses to factors that encourages COVID-19 vaccine acceptance further identified pregnant women as very concerned about the safety of the vaccines. Of all the conditions asked, the responses with the higher percentage were regarding the effectiveness and safety of the vaccines. Also, more pregnant women had a lack of trust in the health system of their countries. In a systematic review, authors found that factors such as trust in the safety and efficacy of vaccines, trust in the individuals that administer or give advice about the vaccines, and trust in the health care system of countries are all important in the vaccine decision-making process [32]. The trust in this study is not far from their lack of confidence in the ability of the health system to appropriately manage their condition when a problem or complication arises due to the deplorable state of most health facilities in Africa and fear of lack of the required competence of health professionals to handle the novel disease. The emergence of COVID 19 exposed the poor conditions of the health systems in terms of infrastructure, equipment, drugs and human resources required for any standard patient care. Additionally, the history of mistrust from the past interactions with official institutions may have influenced the public trust of the participants in this study. Such variable histories and experiences may lead to highly variable and locally specific public trust in vaccines and other immunization programs in the society[33].

A recent study [34] which evaluated the health system functioning in SSA including the challenges and responses has identified the poor structure of health systems and a dearth of essential health services as a major setback during the COVID-19 pandemic. These weaknesses, coupled with the unmet demands arising from the COVID-19 pandemic may have accounted for the pregnant women’s mistrust of the health care systems. Being keen to receive positive feedback from others highlights the need to constantly educate women so they can make informed choices [35]. A detailed vaccine dissemination and outcomes record may also be needed to aid this education.

re point 5
they simply rephrased the statement ever so slightly instead of reviewing a broader literature on side effects, adverse events, media coverage of adverse events, or -- a very easy and appropriate thing to do would be to -- cite recent papers about reasons for covid vaccine hesitancy or refusal. they tried to "soften" their claim about birth defects when that paper - published too long ago - shows they are RARE. I would remove the sentence and reference to that paper, as it is not serving the hypothesis. since so many people in the study mistrust the health system, mistrust the country where the vaccine is made, and are concerned about saftey, the authors need to focus on literature that could explain why they have such low trust. instead, they are spreading reasons to mistrust based on 1 old paper that found birth defects to be rare (a postive vs a negative). 
in the conclusion they state: dispel any misconceptions regarding common false beliefs. but i think they add to the mis-information in this claim.

Response: The sentence on their claim about birth defects when that paper was removed. We have also discussed the mistrust in the health system as advised. See lines 358-363, 376 and 394-400. 400, 429 in conclusion

“The low vaccine acceptance found in the present study may be associated with the safety concerns expressed by the women as most believed in the common myths about the COVID vaccines which significantly influenced the low uptake. This is not different from what was found in other studies especially among Africans where concerns around the safety of the vaccines was the reasons for vaccine hesitancy [30]. Notwithstanding, some side effects have been reported, mostly mild and expected side effects such as pain at site of injection, headache and in some rare cases allergic reactions [31].”

point 2: they claim in the introduction that some women did not have access to covid vaccines yet. but they show no evidence. "they did not assess it" is not an adequate answer. they can easily look at the vaccine rollout of the countries where the survey participants come from and then they know how big or small this issue is. They have the IP address of participants, so while that may not be a perfect match and some may have used VPN, it does give some insights into the availability of the vaccine for participants

Response

We have included a statement in the discussion highlighting that the women may not have had access to the vaccine at the time of this study and included some of the references. The section now reads:

“However, at the time of this study, some African countries had either just rolled out the vaccination program [34, 35] or have only targeted the front-line health workers[36]. “

conclusion should be focused on the aim of the paper and the data. it now seems "off". it does not mention anything about trust building. if people mistrust at the levels they do, why is the conclusion that health promotion officers should educate these women? people don't trust them.

Response: Thanks for this suggestion. We have now included this in the discussion. A revised section in the conclusion now reads:

“More enlightenment campaign should be carried out to create awareness about the safety of the vaccines targeted mainly at high-risk group to drive home the safety and efficacy of the COVID-19 vaccine, as well as dispel any misconceptions regarding common false beliefs. Public health officials can also seize this opportunity to establish meaningful relationships with the communities they serve to gain their trust as this may in turn increase the uptake of COVID vaccination. These approaches should be targeted at those who are married, have tertiary education and people who felt a high-risk perception of contracting the virus”.

this study has great potential to fill a gap in the global literature, as they state is important in the intro (and I agree that it is), yet it falls short in the discussion and conclusions. how is SSA similar or different that other parts of the world when it comes to women (pregnant or not) accepting coivd vaccines?

the fact that many more pregnant women got the covid-19 vaccine  (~30 vs <20%) when compared to the flu vaccine is worth discussing.

Response: This has been discussed as well as the similarity in previous report

“For those women who indicated they did not have access to the vaccine yet, one main reason could be the reduced availability of the vaccines in Africa [27]. Interestingly, it can be seen from this study that more pregnant women took the COVID vaccine compared to the flu vaccine (about 30% vs less than 20%). A similar report was given in a retrospective study [28], where just under 20% women of about 500,000 pregnancies got vaccinated against influenza. The fact that influenza is not easily differentiated from other rampant infectious disease (presenting with fever), such as malaria, which occur in the tropics [29], may have accounted for less attention being paid to this vaccination” [See line 339-346]

Reviewer 2 Report

Dear Authors, please perform a multivariable logistic regression analysis to assess the impact of independent variables on women's acceptance and perception of COVID-19 vaccines. The reason is that in Table 1, several factors affect women's acceptance and perception of COVID-19 vaccines. Thus, matching did not eliminate confounding, and a multivariable model is necessary. 

Author Response

Reviewer 2

Comments and Suggestions for Authors

Dear Authors, please perform a multivariable logistic regression analysis to assess the impact of independent variables on women's acceptance and perception of COVID-19 vaccines. The reason is that in Table 1, several factors affect women's acceptance and perception of COVID-19 vaccines. Thus, matching did not eliminate confounding, and a multivariable model is necessary. 

Response: Thanks for the suggestion.

We have conducted a multivariable analysis on women’s acceptance of COVID vaccination and found that perception was one of the predictors of the model as demonstrated in Table 2 (See abstract line 52-54, line 137, Lines 203-206).

Round 3

Reviewer 1 Report

thank you for addressing the comments. This paper is meanigful and should be published